

# 1 The Future of Conferences

Hazel Gibson[1], Sam Illingworth[2] and Susanne Buiter[3]
1. European Geosciences Union, München, Germany
2. School of Biological Sciences, The University of Western Australia, Australia
3. Division of Earth Sciences and Geography, RWTH Aachen University, Germany
Corresponding author: Hazel Gibson, communications@egu.eu

# 8 Abstract

In the spring of 2020, as the coronavirus swept across the globe, millions of people were
required to make drastic changes to their lives to help contain the impact of the virus.
Among those changes, scientific conferences of every type and size were forced to cancel
or postpone in order to protect public health. Included in these was the European
Geosciences Union (EGU) 2020 General Assembly, an annual conference for Earth,
planetary and space scientists, scheduled to be held in Vienna, Austria, in May 2020. After
a six-week pivot to an online alternative, attendees of the newly designed EGU20: Sharing
Geoscience Online took part in the first geoscience conference of its size to go fully online.
This paper explores the feedback provided by participants following this experimental
conference and identifies four key themes that emerged from analysis of the questions:
what did people miss from a regular meeting; and to what extent did going online impact
the event itself, both in terms of challenges and opportunities? The themes identified are:
connection, engagement, environment, and accessibility; and include discussions of the
value of informal connections and spontaneous scientific discovery during conferences,
the necessity of considering the environmental cost of in-person meetings, and the
opportunities for widening participation in science by investing in accessibility. The
responses in these themes cover both positive and negative experiences of participants
and raise important questions about what conference providers of the future will need to
do to meet the needs of the scientific community in the years following the coronavirus
outbreak.

# 31 1. Introduction


### 33 1.1 The General Assembly of the European Geosciences Union


The European Geosciences Union (EGU) is Europe's leading organisation for Earth,
planetary and space science researchers. Based in Germany, the Union has 18,935



members (as of November 2020) based in more than 135 countries worldwide. Every year
in the spring EGU holds its annual General Assembly in Vienna, Austria; the biggest
geoscience conference in Europe. As a significant part of many Earth, planetary, and
space scientist's research calendars, the EGU General Assembly is a showcase for
research from across 22 Scientific Divisions. The Divisions include fields like
Biogeochemistry, Ocean Science, Atmospheric Science and Solar-Terrestrial science, as
well as more 'traditional' geoscience fields like Geodesy, Geomorphology, Earth
Magnetism and Rock Physics, and Natural Hazards (among many others). In addition to
the scientific research presented, EGU's General Assembly provides researchers with
networking and career development opportunities, training, and the ability to connect with
their extended global community – both personally and professionally. This is especially
key for the Early Career Scientists, who, in 2020, make up 56% of EGU's membership.
At the start of 2020, EGU was seven months into the build-up for the 2020 General
Assembly, which was that year planned to be held from 3-7 May. Apart from the primary
aim of enabling scientists to present their research and learn of the work of their
colleagues, the focus of the 2020 General Assembly as an event was to lie on inclusivity,
accessibility, and environmental sustainability. Inclusivity measures aimed to provide a
safe and respectful environment for all, including the promotion of gender neutral
language, a dedicated person of trust on-site, free childcare, family and breastfeeding
rooms, and a kid's corner. Accessibility measures included dedicated information for
getting to and navigating within the conference centre, wheelchair accessibility, quiet
rooms, catering options, information on visual accessibility, pilots with audio streaming and
auto-captioning, and tips for accessible presenting. Measures aimed at reducing the
environmental impact of the General Assembly centred on travel to and within Vienna,
catering, information sources provided by the EGU, and the conference centre.
Discussions in 2019 and early 2020 involved the consideration of enabling remote
participation, in a manner that would allow remote and on-site participants to directly
engage in questions and discussions.
The annual 'Call for Abstracts' closed in the second week of January 2020 with a new
record of 18,036 abstracts submitted to 701 scientific sessions, compared to the 2019
General Assembly which had 16,273 participating scientists, who presented 16,250 poster,
oral, and PICO (Presenting Interactive Content) presentations in 683 scientific sessions.
By the end of February, the rapidly escalating coronavirus pandemic was the subject of
constant discussion within EGU's governing Council, who began planning several
contingency strategies. By the 19th March it was clear that the conference could not
progress as planned and for the safety of all members it was announced that the in-person
meeting would be cancelled and replaced with an online alternative. However, with less
than six weeks until the start date of the conference, it was also obvious that this could not
possibly be a conference like any previous EGU General Assembly.
**1.2 The 2020 General Assembly: Sharing Geoscience Online**



In designing EGU2020: Sharing Geoscience Online (hereafter EGU20) in the short time
available, the organisers focussed on providing possibilities that could work across time
zones for all authors to present their work and similarly for participants to access the
presentations. To indicate that all presentation formats were equal, previously assigned
poster, oral, or PICO (an interactive presentation form delivered via touch screens)
presentations were turned into a new concept of 'displays'. The decision was made for two
forms of scientific engagement to be possible for each display: pre-uploaded presentation
materials that could be commented on and that were linked to the abstract, and live text-
chat sessions that occurred during the originally scheduled presentation times from the
Programme published on the 9th March 2020 (prior to cancellation). The pre-uploaded
content with comments used EGU's newly launched preprint repository, EGUsphere,
which provided 50MB of storage for participants to upload their presentation using one of
four formats (MP4, JPG, PDF, or PPT). Authors were free to choose what to post with their
abstract, e.g. an animation, a map, a poster, slides, a pre-recorded talk, a pdf and so on.
The uploaded materials were linked to the abstract, which had a DOI, and the materials
were published via open access (in accordance with EGU's publications policy, specifically
a Creative Commons Attribution 4.0 License) unless authors chose a different copyright
statement. The uploads were then made available for comment from the moment they
were uploaded until the 31st of May 2020. Comments and materials remain accessible on
the EGU website (https://meetingorganizer.copernicus.org/egu2020/sessionprogramme)
and EGUsphere (https://www.egusphere.net/conferences/EGU2020/index.html).
The live text-chat function was chosen as a compromise between accessibility and
interaction. Using the host platform 'Sendbird' each of the 701 scientific sessions were
given a text-chat channel that was linked to the pre-uploaded materials of that session and
was moderated by the session conveners (as would be the case for an in-person General
Assembly). There was no limit to the number of people that could digitally attend the live-
text chats and this number varied wildly: though there was a median of 92 participants per
chat, the largest chat had 796 participants. This made for very different experiences of the
text-chat sessions, as the chat window would normally scroll at the speed of the number of
people submitting questions or answers. Participants could also follow multiple chats. EGU
made instructional videos with tips for both conveners and participants that received over
23,000 views by the start of the conference. For example, one of the presenter tips was to
prepare a one or two sentence summary of the display in advance, and this tip was widely
followed.
In addition, some limited online provision had been made for networking and community
building, and there were several live streamed or pre-recorded video sessions – notably
EGU's flagship Union-wide events, the Great Debates and Union Symposia as well as
selected Short Courses. EGU20 brought the annual photo competition online, encouraged
science and art exchanges through the #shareEGUart programme and virtual Artists in
Residence, ran a Data Help Desk, enabled the Division meetings to take place via chat,





and even had an online closing party. The short time that was available to bring the
conference online, however, also meant that other events and activities could not be
scheduled. These included the medal lectures, most of the short courses, most of the
networking events, live-captioning of the Great Debates and Union Symposia, and
measures to help visually impaired scientists (most of whom would not have been able to
participate in the chats). However, as this was nothing like the experience that would
normally be provided to members and the organisers viewed Sharing Geoscience Online
as a pilot since it was the first large Earth, planetary and space science meeting to go
online, EGU's governing Council decided to make attendance free, though only abstracts
that had been submitted by the January deadline were eligible to be presented.
EGU20 launched on the 4th  May 2020 for a week of activities that saw over 22,300
individual users in 721 live text chats who posted approximately 200,400 messages.
11,380 presentation materials were uploaded with the abstracts, which received 6,297
comments by end of the week.
**1.3 Conference feedback survey**
Each year during and after the General Assembly, EGU conducts an online survey among
the participants to ask feedback about the conference experience. The questions consider,
among others, the scientific programme, the role of participants in the conference, and the
additional conference activities, such as Division meetings, the mentoring programme, or
the photo competition. The survey forms an important source of information and feedback
for planning the General Assembly the following year, and have helped to drive positive
change. For example, environmental sustainability and accessibility efforts received extra
support after comments made via these surveys. However, the usual survey, which
assumes among others travel and on-site attendance, was not suitable for Sharing
Geoscience Online.
In order to take advantage of this unique opportunity, as well as to try and gain some
insight into where the potential benefits and challenges of an online conference of this size
may lie, the authors decided to write an entirely new conference feedback survey. Given
the massive upheaval this year it was decided to shorten the usual General Assembly
survey and focus it much more closely on participant experience of this pilot event. The
survey was distributed to all attendees via email and through social media. There were
1,580 complete responses (7% of attendees), which is equivalent to the 2019 response
numbers (n=1,666). Of those complete answers there was a reasonable gender balance
(46% female, 51% male, 0.8% non-binary/other, 3.2% prefer not to say), and 56%
identified as Early Career Scientists. Of the completed surveys, 91.5% said they had never
attended a virtual conference before.



## 2. Methodology

The methodology that was adopted in this study involved surveying participants of EGU20 and asking them for their feedback with regards to their experiences of the online conference. Qualitative content analysis (see e.g., Erlingsson and Brysiewicz, 2017) was then used to interpret the responses to this survey. The questions that were used in this survey can be found in Appendix A. The study was carried out according to the British Educational Research Association's (BERA) ethical guidelines for educational research, ands given that the data contains responses that could lead to the identification of the respondents (even with their name and institute redacted), we have chosen not to make the survey responses available, but a redacted version can be provided upon request.

Any approach which utilises a qualitative content analysis should be guided by the following six steps: formulation of research question; selection of samples to be analysed; definition of categories to be analysed; outline and implementation of coding process; trustworthiness of coding; and analysis of the results of the coding process (Hsieh and Shannon, 2005; Illingworth, 2020). In defining the methodology utilised in this study, we will outline the first five of these steps here, with the sixth (i.e., the analysis) being presented in Sect. 3.

### 2.1 Formulation of research questions

The purpose of this study was to better understand how participants of EGU20 engaged with the online conference, their attitudes in how it compared to a face-to-face event, and whether they thought there were any opportunities that were presented as a result of the event going fully online. This was formalised into the following two research questions (RQs):

RQ1: what did people miss from a regular General Assembly?
RQ2: to what extent did going online impact the event itself, both in terms of challenges and opportunities?

In answering these questions, we are aware that many people's experiences of the conference relate to the technical limitations of the platforms or specific technical issues experienced during the week. Whilst important, we have not addressed those issues in this analysis for two main reasons. Firstly, technical issues and limitations are an issue faced by all types of conference and always impact the experience of the attendee. However, for our specific questions, the exact nature of technical difficulty was not as relevant as the fact that engagement was disrupted. Secondly, it is also important to note the extremely restricted timescale that the organisers had in moving this conference online. As such it is highly unlikely that any scientific conference would be held in exactly this way again – particularly when representing this many people.





## 2.2 Selection of samples to be analysed

The survey was distributed using EGU's preferred survey platform: zohopublic, and the link to the survey was distributed via email to all conveners and authors, as well as EGU members. The link to the survey was also distributed over social media, using EGU's official Twitter, Facebook, LinkedIn, and Instagram accounts, as well as being shared by various other affiliated accounts. The survey was open for responses from the 4th May until the 1st of June 2020.

Once the survey data had been collated and cleaned of incomplete answers, there were 1,580 responses. This entire dataset was used for the initial implementation of the coding process (see Sect. 2.4). Once the initial codes had been set, and in order to more effectively assess the qualitative responses given to the survey, the total dataset of 1,580 responses were divided by career stage (Early Career, Mid-Career or Senior Career) which cumulatively represented 1,503 responses. From these, 50 complete responses that included at least one qualitative answer were selected from each career stage for coding (see Sect. 2.4). This selection included 25 responses from the top of the dataset and 25 from the bottom, representing the first and last respondents to the survey from each career stage, respectively.

## 2.3 Definition of categories to be applied

A conventional approach to qualitative content analysis was adopted in this study, with preconceived categories being avoided, and instead being determined by the implementation of the coding process (see Sect. 2.4). While in some instances a directed content analysis might be more appropriate, this is usually used in those instances where an existing theory would benefit from further description (Hsieh and Shannon, 2005). As the research questions to be addressed in this study are unique, a directed approach is inappropriate. Similarly, a summative content analysis would fail to fully account for the context of the survey responses alongside their content.

## 2.4 Outline and implementation of coding process

To begin with, two of the authors (HG and SI) selected the same random sample of 100 survey responses. We then coded responses to the following survey questions:

- · How effective/timely was EGU at communicating the change to the General Assembly?
- · How would you rate the accessibility of Sharing Geoscience Online for you?
- · How would you rate the technical delivery of Sharing Geoscience Online?





· Was there anything about Sharing Geoscience Online that you would like to see
maintained for future General Assemblies?
· What did you miss most about the General Assembly not being a face-to-face
event?
· What would the ideal format of the EGU General Assembly be according to you?
· In what ways has Sharing Geoscience Online supported / could Sharing
Geoscience Online support your career?
· Any further comments?

The individual codebooks that were used by both HG and SI in this initial coding exercise
are shown in Table 1 and Table 2, respectively. Both HG and SI found that data saturation
had been reached after coding for 100 survey responses, i.e., there were mounting
instances of the same codes, but no new ones.

*Table 1: the codebook that was used by HG in the initial coding exercise, including a*
*definition and an example for each code.*

| Code | Definition | Example |
| --- | --- | --- |
| Networking | Missing in-person interactions, contact, friendship, virtual life | "Seeing my colleagues and interacting in person" |
| Multiple Formats Communicating | Viewing, discussing, listening, debating, multiple format communication | "Verbally communicating to people while visually inspecting their work" |
| Detail | details of science, in depth conversation | "Without the visual interface it's very difficult to go into details" |
| Behaviour | people do not have respect, people are angry, stressed, rude | "people don't respect their time slots and have cross conversations" |





| Spontaneity | Missing freedom within schedule, time to talk, debate, explain, find unexpected subjects, interactions or conversations | "spontaneous questions, time for a more personal, friendly chat" |
|---|---|---|
| Preparation | Preparation of scientific materials, talks, formats etc | "scientifically I could prepare/have more in depth discussion" |
| Flexibility | Flexible interactions, being able to move between sessions, multi-tasking | "often the whole session is not totally of interest and you would like t change room just for one talk" |
| Open Access Science | open access science, sharing science, expands reach of research | "the impact is undoubtable greater than in classic EGU GA where only a few people could stand in front of poster" |
| Emotion / Nostalgia | Missing the whole event, an intangible sadness, non-specific, excitement and joy, boredom | "Everything! Nothing can replace the face-to-face event" |
| Overcoming Current Events | Overcoming non-specific challenges of coronavirus to carry on with plans | "You did an amazing job in a short time, and considering the current situation in the world" |





| Attendance | Able to attend or not attend meeting despite original plans | "it has allowed me to attend a meeting I could not attend in the first place" |
|---|---|---|
| Waste of time | it was a waste of time and disappointment, better off cancelling | "I don't see the point of this format, EGU had better been completely cancelled" |

*Table 2: the codebook that was used by SI in the initial coding exercise, including a*
*definition and an example for each code.*

| Code | Definition | Example |
|---|---|---|
| Deeper engagement | These responses indicate that these participants were able to have a deeper engagement in terms of either more questions or longer discussions etc. | "Scientifically i could prepare/have more in-depth discussion." |
| Good for Early Career Scientists | Presented good opportunities for Early Career Scientists. | "During oral presentations, generally time for questions is very narrow, and you do not always feel it is your place to do so as an ECR. Having this ability during the whole session time slot is really enjoyable." |
| Difficulties with Tech | Participants encountered difficulties accessing the online content. | "The chat pages has some glitches. Comments sometimes disappearing for unknown reasons in my window, while other people could see them." |





| | | |
|---|---|---|
| Networking | Participants missed the opportunity to professional network in person. | "Meeting people! Networking!  The chat it great but it is just not the same." |
| Socialising | Participants missed the opportunity to catch up with old colleagues and friends in person. | I can't see my teachers and classmates, we can't talk questions face to face, sometimes ,the text-chat can't arrive the effect. And I miss the scenery and food of Austria, haha. |
| Too much info | Participants felt overwhelmed with the amount of comms they received. | "The emails where too long and un-structured, plus a bit spammy (emails as author, co-author, personal program, convener....)" |
| Lack of engagement | These responses indicate that the online format presented less opportunities for deep engagement on scientific topics. | "The 15-min orals and as long as need discussion for the posters. This format cuts down on the ability to explain, drastically. I don't think it's been translated good enough." |
| Environment | Attending the conference online had a positive impact on the environment. | "carbon footprint issue. Obviously we do not need to go every year to such meetings. So remotely following them is very interesting. And if you have personal restrictions (accessibility, money, child care) preventing you to attend, that's quite an improvement!" |





| Boring | The online event was less vibrant than the face-to-face meeting | "Nothing special and there are plenty of ways to explore to make this feel more interactive. Scrolling through the presentations makes attendance feel a lot like grading papers." |
|---|---|---|
| Convenience | The online event was more convenient to attend. | "Reduce long distance transportation while maintaining the visual and verbal aspects" |
| Lack of info | Difficult for people to 'discover' the conference or find out how to attend specific webinars etc. | "Found it hard to access the talks or find info about how to attend webinars but the rest was well advertised" |
| Inaccessible | The online format proved inaccessible to some people. | "I can't concentrate on the virtual meeting, although it's great, especially in text-chat section, I can't follow other people's idea." |
| Accessible | The online format proved to be more accessible for some people | "Those unable to physically attend can gain some part of the experience from home. That includes physically disabled and financially unable." |
| Discovery | Online events less likely to have the 'accidental discoveries' possible in the physical version | "Meeting up with friends, meeting new people, walking around, randomly finding interesting sessions" |


After this initial coding exercise was completed, HG and SI combined their codebooks and
decided on a number of categories that covered all of these codes, and which could be
used to better represent the themes that were emerging from the data. These combined
categories are shown in Table 3.

*Table 3: the initial combined categories that were used to classify the initial codes of HG*
*and SI.*



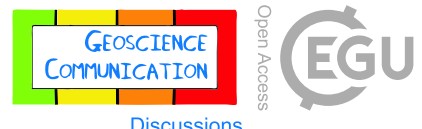

| Category | Definition | Codes (Original Coder in brackets) |
|---|---|---|
| Information | How participants were informed of the new format, and how they accessed this information. | Attendance (HG), Waste of Time (HG), Difficulties with Tech (SI), Too much Info (SI), Lack of Info (SI) |
| Connecting | How networking and socialising were impacted by moving to a virtual conference. | Networking (HG), Networking (SI), Socialising (SI) |
| Engagement | The extent to which the online environment either encouraged or restricted engagement. Also includes spontaneity / discovery of sessions. | Multiple Format Communicating (HG), Spontaneity (HG), Preparation(HG), Emotion / Nostalgia (HG), Deeper Engagement (SI), Lack of Engagement (SI), Boring (SI), Discovery (SI) |
| Environmental Impact | How changes to an online conference impacted the environment. | Overcoming Current Events (HG), Environment (SI) |



| Accessibility | The extent to which an online conference was more or less accessible to different audiences. | Detail (HG), Behaviour (HG), Flexibility (HG), Open Access Science (HG), Convenience (SI), Inaccessible (SI), Accessible (SI) |
|---|---|---|
| Early Career Scientists | The impact that the online environment had on Early Career Scientists. | Good for Early Career Scientists (SI), |

After these combined categories had been determined, both HG and SI re-visited the
original RQs and decided that some of the survey's questions, whose responses had been
analysed in the initial coding exercise, were not related to these RQs. The following
questions were selected as being most pertinent to answering the RQs (given in
parentheses) of this study:
·   How would you rate the accessibility of Sharing Geoscience Online for you?
(RQ1)
·   Was there anything about Sharing Geoscience Online that you would like to see
maintained for future General Assemblies? (RQ2)
·   What did you miss most about the General Assembly not being a face-to-face
event? (RQ2)
·   What would the ideal format of the EGU General Assembly be according to you?
(RQ1, RQ2)
·   In what ways has Sharing Geoscience Online supported / could Sharing
Geoscience Online support your career? (RQ2)
·   Any further comments? (RQ1, RQ2)
The other questions (i.e., 'How effective/timely was EGU at communicating the change to
the General Assembly?' and 'How would you rate the technical delivery of Sharing
Geoscience Online?) were deemed to be more related to the technical delivery of an
online conference rather than specific learnings and attitudes towards the experience of a
face-to-face or online event. At this stage in the analysis, the data was cleaned up to
remove any responses that did not contain information, and also to split the respondents



into three broad categories: Early Career Scientists, Mid-Career Scientists and Senior
Career Scientists. This split was done according to the specific information that had been
provided by the respondents, who as part of the survey ('What is your career stage /
employment status?') had to self-identify as to which of these categories they belonged to.
After cleaning the data, the categories shown in Table 3 were again revisited, and it was
decided that the 'Information' and 'Early Career Scientists' categories should be dropped
from the subsequent analysis. The former because of the same reason outlined for
neglecting two of survey questions in this stage of the analysis (i.e., because the
responses were more concerned with technical changes and difficulties), and the latter
because it was decided that it would be discriminatory to highlight one of the three groups
of researchers. As a result, the categories that are shown in Table 4 are those that were
used for this final stage of coding and analysis.
*Table 4: the final categories that were used in the analysis of the responses to the survey.*

| Category | Definition |
| --- | --- |
| Connecting | How networking and socialising were impacted by moving to a virtual conference. |
| Engagement | The extent to which the online environment either encouraged or restricted engagement. Also includes spontaneity / discovery of sessions. |
| Environmental Impact | How changes to an online conference impacted the environment. |
| Accessibility | The extent to which an online conference was more or less accessible to different audiences. |

For the final stage of coding, 50 random respondents from each of the three distinct
demographic groups (i.e., Early Career, Mid-Career, and Senior Career) were selected.
HG and SI then individually assigned the categories shown in Table 4 to the responses to
the questions given above for these respondents. Both HG and SI observed that for each
of these 50 sets of responses, the categories that are shown in Table 4 could be assigned,





with no newly emergent codes or categories during this process, therefore providing
confidence that the categories shown in Table 4 were the dominant themes to emerge
from the data, and which will be discussed further in Sect. 3.

**2.5 Trustworthiness of coding**



At each stage of the qualitative content analysis that was adopted in this study, the
individual codes and categories were re-examined in order to confirm that they accurately
captured the responses of the survey in relation to the RQ. Both HG and SI carried out this
coding independently, until there were no further codes or categories found to be emerging
from the data, i.e., until descriptive saturation had been reached (Lambert and Lambert,
2012). Similarly, a combination of systematic sampling, constant comparison, and proper
audit and documentation (see Sect. 2.2 and 2.4) were used to ensure both the reliability
(i.e., the consistency with which this analysis would produce the same results if repeated)
and the validity (i.e. the accuracy or correctness of the findings) of this approach (Leung,
2015).

# 3. Results & Discussion

As can be seen from Table 4, four major categories emerged from the methodology that
was adopted in analysing the responses to the survey. We now discuss each of these
emergent categories, how they relate to RQ1 ('What did people miss from a regular
General Assembly?') and RQ2 ('To what extent did going online impact the event itself,
both in terms of challenges and opportunities?'), and how they compare to other research
that has been conducted in terms of the transitioning of large academic conferences from
physical to virtual spaces.

## 3.1 Connecting


One of the categories identified from the responses from attendees of EGU20 was
'connecting'. This was defined as the interpersonal connections between attendees of the
conference; the human-to-human, individual, or informal interactions. This category is
distinct from the connections made around the scientific content, which is discussed in
'engagement' (Sec. 3.2).

The responses coded in this category were frequently posted in direct response to the
survey question 'What did you miss most about the General Assembly not being a face-to-
face event?', and the responses were most often framed as negative or expressing loss. In
general, the descriptions of the loss of connection during EGU20 can be summarised as
being those opportunities to interact with colleagues and friends 'beyond the session'.  The





loss of connection was most often described in terms of informal interaction, such as this
observation from a Senior Career Scientist:

"Personal communications. The possibility to share a lunch or a dinner together with
potential future colleagues."


Networking was also a key aspect of the loss of connection, particularly expressed by Mid-
Career Scientists and Early Career Scientists searching for career development. The
limited scope of a platform such as the one that was provided during EGU20 for
networking, echoes findings of other studies, wherein social media and other digital
platforms are often used to build networking potential, which is then followed up for more
meaningful discussion in-person (Reinhardt et al, 2009; Kimmons and Veletsianos, 2016).
The discussion of a loss of connection in networking was also described as a function of
learning who is potentially a valuable contact and meeting new people, as this Mid-Career
Scientist observed:

"The ability to network. Randomly meet people you don't even think you're
interested in meeting."


The loss of connection for Senior Career Scientists was especially pronounced in the way
they described friendship and treasured colleagues. This was not, however, limited to
Senior Career Scientists, and often included an aspect of nostalgia for the conference
itself and an enjoyment of the city of Vienna. Many respondents described the loss of
contact with friends as central to their General Assembly experience, as this Senior Career
Scientist responded:

"90% of my motivation to go to the EGU General Assembly is to meet with
colleagues and friends in person. That's a great loss."


The final aspect of loss with regards to the theme of connection was in the stimulus and
inspiration that comes from informal conversation and meetings with people. This was
expressed in the form of being able to plan future activities, come up with new ideas, or
simply the inspiration that breaking the routine through connection provides, as this Early
Career Scientist describes:

"Networking, meeting people in person, the atmosphere of the meeting, Vienna, and
listening more than reading. My job as a scientist is mostly reading and writing, the
physical conference is breaking out of this, which opens many other opportunities to
think, cooperate, and pathways to discuss."


These responses show that though the scientific content is key to any conference, the
ability to build and experience meaningful informal connections with friends and colleagues
for both personal and professional reasons, is very valuable to attendees, which is



something that is also present in studies of remote working more generally (Nardi and
Whittaker, 2002). This aspect of proving space 'beyond the session' for informal interaction
is a useful recommendation for face-to-face conferences as well, but for digital or online
conferences may provide critical to their success or failure.

## 3.2 Engagement


Another category to arise from the responses from respondents was that of 'engagement'.
Specifically, this was related to the extent to which respondents were or were not able to
engage with both the online format and the material that was presented.

In terms of criticisms, several respondents felt as though the format of EGU20 precluded
the depth of conversation and scientific rigour that would normally be expected at the
conference, as demonstrated by this comment from a Senior Career Scientist:

"Maybe I come from an old school, but attending a conference directly offers many
possibilities to establish contacts with other scientists, to interact in a deeper and
less aseptic way than online event provides."

However, others actually found more opportunity for engagement, both during and after
the various sessions. For example, one Early Career Scientist observed that:

"It may be topic related, but this time was the first time that I got exactly the kind of
feedback to my presentation I was hoping for. And that came one-two days after the
actual presentation via the discussion section and via email."

This dichotomy of opinions was observed across all three respondent groups, and a
similarly polarising aspect of engagement was the spontaneity of discovery that is
associated with large conferences like the EGU General Assembly. Some respondents
noted that one of the things they missed the most was the opportunity to accidentally or
purposefully walk in on sessions outside of their field of expertise, thereby helping to
cross-pollinate scientific discourse and helping them to develop their own interdisciplinary
approaches. This attitude is evident in the following comment from a Mid-Career Scientist
when noting what it was that they missed most about EGU20 not being a face-to-face
event:

"Wandering around and going to attend a random session outside of my field of
expertise."



However, others felt the exact opposite, i.e. that the online format actually made it more
possible to engage in research outside of their specific field of expertise, as evidenced by
this comment from a Senior Career Scientist:
"I could take part in sessions at the fringe of my expertise since the short
summaries given by presenters helped me to understand their core message."
The 'short summaries' that this respondent refers to, in combination with the pre-uploaded
longer presentations, is one facet of engagement that seems to have been received with
almost unanimous positivity. For EGU20's scientific sessions, authors were encouraged to
upload and share their presentation materials and opt in to commenting from 1 April 2020
onwards, and then prepare a one or two sentence summary of these presentation
materials for the live text chat. This meant that participants had up to a month to view other
researchers work in detail and prepare any questions for the allocated session and
associate chat during the week of EGU20 itself (4 to 8 May 2020). The opportunity to view
this work in advance was a frequent feature of responses to the question 'Was there
anything about Sharing Geoscience Online that you would like to see maintained for future
General Assemblies?'. For example, one Early Career Scientist noted that:
"This made it much easier to think about the contents without the stress of
everything around you in the conference centre."
The following comment from a Mid-Career Scientist echoed the sentiment of many
respondents that this is a feature that should be utilised in future General Assemblies:
"Uploading "displays" online, for anyone to see and comment. Even for a physical
meeting it would be useful for the general public, or the colleagues who couldn't
make it (either to the conference or to the session)."
However, the positive response to this pre-release of information must be caveated by the
concerns that many respondents raised around potential issues with intellectual property
and the dangers of permanently hosting preliminary results online, as evidenced by the
following comment from a Mid-Career Scientist:
"I'm concerned about the copyright issues when uploading presentation."
One Senior Career Scientist went further, noting that:
"Conferences are often about discussing preliminary results, when I submit an
abstract I DO NOT subscribe to permanently DOI-ing preliminary results."
The outcomes of this category are very mixed, with some respondents finding EGU20 to
be less engaging than a normal General Assembly, whilst others noted that it actually



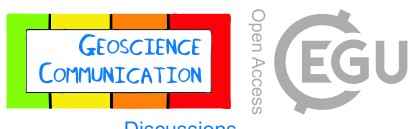

presented more opportunities for deep engagement. It would appear that attitudes towards
engagement depend very much on the respondent's personal attitudes towards online vs.
face-to-face conferences, and a more general comment is that EGU20 does not appear to
have swayed many respondents from what are clearly deeply entrenched viewpoints. One
thing that is made clear from the respondents, however, is that they deeply valued the
opportunity to view scientific research in advance of the conference, although this option
needs careful consideration with regards to intellectual property and the sharing of
preliminary results.
## 3.3 Environmental Impact
One of the clear opportunities that arose from the EGU20 format was the positive impact
that this was perceived to have on the environment, i.e., through the reduced carbon
emissions associated with attendees travelling to Vienna to participate in a General
Assembly. This manifested itself across all three distinct demographic groups (Early
Career Scientist, Mid-Career Scientist, and Senior Career Scientist).
EGU has previously taken several steps to mitigate and offset the impact that travel to the
General Assembly has on the environment. These include: giving participants the
opportunity to offset the $CO_2$ emissions resulting from their travel to and from Vienna (in
2018 and 2019, voluntary carbon offsetting through EGU was used by 25% to 32% of
attendees), advising participants to travel by train to Vienna when possible (and promoting
discounts offered by train companies to participants); and encouraging participants to use
public transportation once in Vienna, by giving away a weekly transportation pass with
every week ticket to the conference.
Of course, the environmental impact of hosting a large conference like the EGU General
Assembly extends beyond that of travel, and also includes the printing of materials, the
consumption of power at the venue, and the sourcing of catering. The conference venue,
the Austria Centre Vienna, has a number of green measures in place, including having
energy-saving LEDs throughout the centre, using a solar array to heat the water used in
the kitchens and toilets, and working with an in-house catering company compliant with
green standards. Other measures that have been implemented to reduce the
environmental impact of the General Assembly include no longer offering single-use water
bottles during breaks, installed water fountains for refilling multi-use bottles, phasing out
printed copies of the programme book, and making sure that the lanyards are created out
of 100% recyclable materials. If the 2020 event had taken place in Vienna, all travel of
participants would have been carbon offset, a start would have been in phasing out single-
use coffee cups, and bicycle transport in Vienna would have been promoted. However,
from the results of this survey, these steps do not go far enough to alleviate the concern
that many of the respondents have with regards to the environmental impact of the
General Assembly. Furthermore, as noted by Hischier and Hilty (2002), the environmental



impact of a large international conference such as the EGU General Assembly is
dominated by the travel activities of the participants. Here long-range flights are the
dominant element, as exemplified for the 2019 Fall Meeting of the American Geophysical
Union where 75% of the emissions were due to intercontinental flights over distances
larger than 8,000 km made by 36% of the attendees (Klöwer *et al.*, 2020). Klöwer points
out that for the 2019 EGU General Assembly in Vienna, Virtual participation for 26% of the
highest emitting participants would reduce the carbon footprint by 80%
(https://github.com/milankl/CarbonFootprintEGU). As such, despite any green measures
that EGU may take in Vienna, minimizing air travel is the only way to ensure a significant
reduction in environmental impact.
The hard decisions that many researchers face with regards to the environmental impact
of attending the General Assembly are evident from the following two comments (both
from Early Career Scientists):

"As geologists we really need to think about being more climate-friendly in our jobs!"

And
"In order to cut the carbon footprint of science, we need to go online more and have
less [SIC] actual meetings (although I prefer those)"

Despite these quotes coming from Early Career Scientists, this environmental conflict of
interest was felt keenly across the three groups. For example, one Senior Career Scientist
observed that:

"…because the environmental foot print [SIC] of normal EGU seems unreasonable
nowadays, we have to think differently and this crisis pushes a bit to (*SIC*) far but
shows us alternatives."

As a result of this conflict of interest, many of the respondents (across all three groups)
suggested varying hybrid models of face-to-face and online options for future EGU
General Assemblies, citing environmental concerns as their primary reasons for moving
away from a strictly 'business as usual' model.
The internal conflict of several of the respondents is appropriately reflected by this
comment from a Senior Career Scientist:

"The online format is a great opportunity to reduce the environmental impact of the
GA [General Assembly] and allows people to attend who cannot travel. But face to
face meetings are important too. I would favour alternating between online and
physical meetings. [SIC] in the future. Both have advantages."



16,273 scientists participated in the EGU General Assembly 2019 in Vienna, Austria.
Klöwer *et al*. (https://github.com/milankl/CarbonFootprintEGU) estimated that these
scientists travelled in total 94 million km to Vienna and back, which emitted 22,300 tonnes
of carbon dioxide equivalent (tCO$_2$e) , an average of approximately 1.4 tCO$_2$e per scientist
To put this into context, this is the total weekly carbon footprint of approximately 27,000
average American households, and based on other studies (see e.g., Green, 2008; Jäckle,
2019; Bousema et al., 2020), this might be considered to be a conservative estimate.
As noted by Bousema et al. (2020), although in-person meetings have many benefits, the
ecological impact of conference travel is considerable and demands action. With more
than 16,000 attendees the EGU General Assembly has a substantial environmental impact
and whilst the EGU has taken several steps to reduce their impact, it is clear that this is an
issue that is not being adequately addressed. Even allowing for the environmental impact
of hosting a large online event (Versteijlen et al., 2017), the reduction in carbon emissions
from thousands of people not travelling to Vienna every year is substantial. Whatever
format is taken by future EGU General Assemblies, the results of this survey indicate that
something needs to be done to better mitigate the environmental damage that a face-to-
face conference presents in its current guise. Perhaps this is the opportunity we have been
waiting for to lead by example and transition to a General Assembly that not only presents
research on how to mitigate climate change, but also takes actionable steps in doing so.
As observed by one Early Career Scientist:
"If it was only online, we'd have to adapt to a new way of working, which would
ultimately accelerate our transition to a green future"

## 593  3.4 Accessibility

The fourth category identified in coding is one that is often cited in connection with the
benefits of online conferences: 'accessibility'. In this case accessibility was related to any
discussion of increasing the ability of people to participate in the General Assembly,
regardless of the reason for their inability to participate at other times. Though this has
particular relevance to under-represented groups in academia, such as those who have a
disability, caring responsibilities, financial constraints or are excluded due to systemic
oppression, this category also included people who may attend in a normal year, but
couldn't for a specific reason in 2020.
The first thing to note here is that responses coded as being about accessibility were
overwhelmingly positive. There was a general appreciation of the ability for an online
General Assembly to widen participation – particularly for those who would not normally be
able to attend as these Early Career Scientists stated:
"Those unable to physically attend can gain some part of the experience from
home. That includes physically disabled and financially unable."



And:
613  "I think the online format allowed people who could not come to the meeting for cost
614  or travel restrictions to attend, thus broadening the scientific content."
Financial constraints were often stated as a limiting factor, but connected to this was the
burden of travel and all that it entailed – particularly the challenge of obtaining
documentation for residents of certain countries – but many also recognised the value of
being able to invite non-traditional conference attendees that would also normally
experience a financial barrier, thus encouraging open science, as this Mid-Career Scientist
stated:
623  "Open access and open chat to everyone who can log in with their email; also
624  stakeholders could attend as a guest!"
In addition to improving the accessibility of the scientific information, there was also note
made of improving the accessibility of the format to support those less inclined to engage
in traditional forms of conference questioning (which can be quite combative at times) to
people who are perhaps at an earlier career stage, or of a more introverted personality, as
observed by this Mid-Career Scientist:
632  "Accessibility for those with caring responsibilities, lack of financial resources, etc.
633  And the fact that many are more comfortable asking questions in an online format >
634  good for introverts and ECRs."
However, many stated that despite the improved accessibility, the online conference was
something that should in future be relegated to being supplemental to a traditional in-
person conference. Some even described the accessibility of an online conference as a
trade-off, as this Senior Career Scientist said:
641  "The expanded attendance is good, but there is definitely something lost: but also
642  something gained (accessibility)."
The benefits of an online conference for accessibility cannot be ignored, and it's important
to note how many respondents also identified ways in which accessibility in this regard
truly went beyond some narrower definitions to real widening participation. As with other
discussions of accessibility, questions remain as to who is included in this survey and who
is excluded, and how online engagement continues to include or exclude certain people,
often compounding exclusion in non-digital spaces (Khalid and Pedersen, 2016).




# 4. Conclusion



The original purpose of this study was to address the following two research questions:

RQ1: what did people miss from a regular General Assembly?
RQ2: to what extent did going online impact the event itself, both in terms of challenges
and opportunities?

As can be seen from Sect. 3, it is evident that there are several aspects of a face-to-face
EGU General Assembly that were missed by respondents, not least the opportunity to
connect and interact with colleagues in informal environments. It is also clear from these
emergent themes that there are many aspects of going online that present opportunities
that should not be forgotten for future General Assemblies. The future of the EGU General
Assembly is something that requires careful consideration, and indeed many of the
choices are driven by change outside the control of the EGU Executive and Programme
Committee; the 2021 General Assembly has already been announced as being a fully
online event because of the restrictions that continue to be imposed by the coronavirus.
However, there are still many variables that are within their control, and it is clear from the
responses to the survey that many participants feel very strongly that a fully online, or
hybrid General Assembly is not only an option but a necessity, in order to both make the
conference more accessible and also to address the significant environmental impact of
hosting a face-to-face intentional conference. In moving towards any digital provision for
future General Assemblies, we would like to offer the following recommendations, which
have emerged from the results of this study:

1. **The online provision should not just be an afterthought.** An online digital
conference cannot simply be a replication of a face-to-face version. Similarly, if a
hybrid option is pursued, then there needs to be equal value attached to both
the face-to-face and digital aspects. Care should be taken to enable direct
interactions between those on-site and remote participants.
2. **There needs to be an accessible and innovative space to enable informal**
**connections.** One of the biggest issues that needs to be addressed in an online
environment is in creating spaces where researchers can meet up with old
colleagues, encounter new ones, and informally engage with one another. The
café culture of Vienna cannot be replicated in an online format, but then nor is it
replicated in the actual General Assembly itself. Digital interactions that take
place on platforms that already exist for such encounters need to be considered.
3. **Accessibility needs to be re-considered.** Online conferences make science
much more accessible to many different groups and helps to truly diversify
science. However, it also presents several additional access needs that need to
be considered. These include, but are not limited to: digital literacy, accessibility





for visual or hearing impaired participants, access to fast and reliable
broadband, and limitations imposed by time zones.

**4. The sharing of preliminary results needs to be carefully thought through.**
One of the highlights from EGU20 was the capacity for people to see (and
comment on) scientific research before it was presented. Enabling this feature
for a future General Assembly would be well-received, but careful consideration
needs to be given as to how to ensure that all researchers feel confident that
their research is protected as we increasingly move into an era of Open
Science, especially for those who work with confidential data.


The validity and reliability of this study is discussed in Sect. 2.5, but it should be noted that
as with any qualitative analysis there is a degree of interpretation in the analysis of the
responses to the survey. However, we are confident that the emergent categories are
representative of the general zeitgeist of EGU participants.

The format of EGU20 was radically changed because of the impacts of the coronavirus,
and whilst there are clearly issues that need to be addressed for any future online version
of the EGU General Assembly (either fully online or in some hybrid form), it has perhaps
forced a change that might not have otherwise occurred. The organisers and participants
of subsequent General Assemblies need to think very carefully about whether the
perceived positive impacts of a traditional face-to-face conference outweigh the very real
concerns about inclusion and environmental impact. Or as one of the respondents to the
survey noted:

"The traditional conference is getting more difficult to justify with climate change and
the requirement that everyone jet around the world to discuss earth science,
especially science related to climate change."


If the community does not listen to these requests and consider them very seriously, then
we are at risk of being nothing more than a data point on the 'business-as-usual' climate
simulations that many of us have dedicated our professional lives to avoiding occurring at
all costs.

# 724    Data availability


Given that the data contains responses that could lead to the identification of the respondents
(even with their name and institute redacted), we have chosen not to make the survey responses
available, but a redacted version can be provided upon request.



# Competing interests

Author Hazel Gibson is an Associate Editor of *Geoscience Communication*, Author Sam Illingworth is the Chief Executive Editor of *Geoscience Communication*, Author Susanne Buiter was the chair of the Programme Committee for EGU2020: Sharing Geoscience Online and is Executive Editor of *Solid Earth*.

# Acknowledgements

The authors thank the hundreds of volunteers around the globe who have worked so hard to shape EGU2020: Sharing Geoscience Online, an exciting experiment in response to the COVID-19 pandemic and a great success throughout the entire week, and especially the Programme Committee (Raffaele Albano, Jonathan Bamber, Anouk Beniest, Johannes Böhm, Marc De Batist, Ira Didenkulova, Michael Dietze, Olaf Eisen, Fabio Florindo, Helen M. Glaves, Karen Heywood, Marian Holness, Patric Jacobs, Philippe Jousset, Chris King, Olga Malandraki, Mioara Mandea, Sonja Martens, Alberto Montanari, Athanasios Nenes, Lena Noack, Lara Pajewski, Giuliana Panieri, Dan Parsons, Maria-Helena Ramos, Didier Roche, Claudio Rosenberg, Håkan Svedhem, Paul Tackley, Peter van der Beek, Stéphane Vannitsem, Stephanie C. Werner, Claudio Zaccone) for their tireless efforts. The authors would also like to extend their thanks to Copernicus Meetings (Mario Ebel, Katja Gänger, Katharina Huckemeyer, Katrin Krüger, Martin Rasmussen, Stefan Schwardt and Hennadii Shvedko) and to the other EGU Office staff (Terri Cook, Chloe Hill and Philippe Courtial) for their dedication to making Sharing Geoscience Online a success.

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

# Appendix A:
**EGU Sharing Geoscience Online 2020 survey questions.**

*Thank you for participating in the feedback survey for EGU Sharing Geoscience Online 2020! This*
*has been an unprecedented experiment, where we organised the largest virtual gathering of*
*geoscientists ever, in only 6 weeks since the cancellation of the physical General Assembly. We*
*are very curious about your experience at Sharing Geoscience Online: what has worked well, what*
*could be better, what did you miss, and what should EGU consider to keep for future meetings.*

*We would like to ask you to take 5-10 minutes to complete this questionnaire, as your input is very*
*helpful for shaping future EGU General Assemblies and possible virtual extensions.*

*Susanne Buiter (RWTH Aachen University)*
*Chair of the EGU General Assembly 2020 Programme Committee*

Q1. What EGU programme groups do you associate most closely with?
– Atmospheric Sciences
– Biogeosciences
– Climate: Past, Present & Future
– Cryospheric Sciences
– Education and Outreach Sessions
– Earth Magnetism & Rock Physics
– Energy, Resources & the Environment
– Earth & Space Science Informatics
– Geodesy
– Geodynamics
– Geosciences Instrumentation & Data Systems
– Geomorphology
– Geochemistry, Mineralogy, Petrology & Volcanology



| 810 | – | Hydrological Sciences |
|---|---|---|
| 811 | – | Interdisciplinary & Transdisciplinary Sessions |
| 812 | – | Natural Hazards |
| 813 | – | Nonlinear Processes in Geosciences |
| 814 | – | Ocean Sciences |
| 815 | – | Planetary & Solar System Sciences |
| 816 | – | Short Courses |
| 817 | – | Seismology |
| 818 | – | Special Scientific Events |
| 819 | – | Stratigraphy, Sedimentology & Palaeontology |
| 820 | – | Soil System Sciences |
| 821 | – | Solar-Terrestrial Sciences |
| 822 | – | Tectonics & Structural Geology |
| 823 | – | None |
| 824 | | |
| 825 | Q2. What is your present country of employment / study? | |
| 826 | | |
| 827 | Q3. What is your gender? | |
| 828 | – | Female |
| 829 | – | Male |
| 830 | – | Non-Binary |
| 831 | – | Prefer not to say |
| 832 | – | Prefer to self describe |
| 833 | | |
| 834 | Q4. Did you feel restricted to participate in the conference due to some physical limitations? | |
| 835 | | |
| 836 | Q5. Does any of the following apply? | |
| 837 | – | It is difficult for me to attend physical meetings, but I could attend Sharing Geoscience |
| 838 | | Online |
| 839 | – | It is difficult for me to attend physical meetings and I also experienced difficulties attending |
| 840 | | Sharing Geoscience Online |
| 841 | – | I can attend physical meetings, but experienced difficulties attending Sharing Geoscience |
| 842 | | Online |
| 843 | – | I can attend physical meetings and Sharing Geoscience Online |
| 844 | – | Other / Comments |
| 845 | | |
| 846 | Q6. Why did you give this answer? | |
| 847 | | |
| 848 | Q7. What is your career stage / employment status? | |
| 849 | – | Early career scientist |
| 850 | – | Mid-career scientist |
| 851 | – | Senior scientist |
| 852 | – | Retired |
| 853 | – | Self-employed |
| 854 | – | Not currently employed |





– Other
Q8. What is your role at EGU Sharing Geoscience Online 2020?
(Tick all that apply)
– Abstract author or co-author
– Session convener or co-convener
– Session chair
– EGU division scientific officer
– EGU Programme Committee member
– EGU council member
– Scientific participant
– Press/media
– Other (Please State)
Q9. Have you attended a virtual conference before?
Q10. Which one?
Q11. How effective/timely was EGU at communicating the change to the General Assembly?
– Very Good
– Good
– Average
– Poor
– Very Poor
Q12. Why did you give this score?
Q13. What were your main sources of information about the changes to the General Assembly?
(Tick all that apply)
– EGU website (www.egu.eu)
– General Assembly website (www.egu2020.eu)
– Social Media
– Blogs
– Newsletter
– E-mails by EGU/Copernicus
– Other (Please specify)
Q14. Which activities of Sharing Geoscience Online did you participate in?
– Scientific Sessions
– Union Symposia
– Great Debates
– Short Courses
– Townhall Meetings
– Photo Competition
– #shareEGUart





–    Division Meetings
–    Networking Events
–    Closing Party
Q15. How many different chat sessions of Sharing Geoscience Online did you participate in?
Q16. How would you rate the accessibility of Sharing Geoscience Online for you?
–    Very Good
–    Good
–    Average
–    Poor
–    Very Poor
Q17. Why did you give this answer?
Q18. How would you rate the technical delivery of Sharing Geoscience Online?
–    Very Good
–    Good
–    Average
–    Poor
–    Very Poor
Q19. Why did you give this answer?
Q20. Was there anything about Sharing Geoscience Online that you would like to see maintained
for future General Assemblies?
Q21. What did you miss most about the General Assembly not being a face-to-face event?
Q22. What would the ideal format of the EGU General Assembly be according to you?
–    Face-to-face event only
–    Mixed face-to-face and online event
–    Online event only
Q23. Why did you give this answer?
Q24. In what ways has Sharing Geoscience Online supported / could Sharing Geoscience Online
support your career?
Q25. Any further comments?