# Peer review of "The Future of Conferences: lessons from Europe's largest online geoscience conference"

_Geoscience Communication, 2020_

## Author Comment (AC1)

Reviewer 1 Response

*I really enjoyed reading this paper. The topic is one that will be of interest to any geoscientist or scientist today. The survey captured the views of scientific audiences at a unique moment in time, when virtual conferences had not yet become the norm (91.5% of participants said they had never attended a virtual conference before). The qualitative analysis employed in this paper is entirely appropriate. The paper is well written and the method is clearly explained, with subject-specific technical terms defined (e.g., lines 259-260 and Section 2.5). The quotes selected in the paper are engaging and appropriate to the themes. The results and conclusions clearly follow from the analysis, and the ending of the paper is poignant (lines 706-722). I particularly appreciated the self-reflective tone of the paper and think this work brings up some important issues for geoscientists to reflect upon.*

*My main suggestion to the authors is to ensure it is clear, especially in the title, abstract and conclusions, what the results mean for the EGU assembly, and what they might mean for other future conferences. The title of the paper is very effective in that it immediately draws the reader in, but it also seems to promise that a solution will be offered in the paper. What might the future of conferences look like? Can EGU's experience help light the way ahead for others? The recommendations provided at the end of the paper (line 672) are stated as valid for 'future general assemblies', but are some also likely to be valid for conferences in general? I suggest that to elaborate on this question (limitations of the sample in mind) would be useful in terms of aiding the scientific community in organising future events.*

Dear Anthea,

Thank you for this feedback to our paper, we are really pleased that you enjoyed it and saw the relevance of the work. We also appreciate the constructive feedback you provided, and we will address your main suggestion here and then the minor notes below. Firstly, with regard to the scope of the paper as suggested in the title, abstract and conclusions – we very much appreciate the question about broadening the scope of the paper to reflect on other geoscience conferences, but, as you mention, we are also limited by the sample and survey design. Thus, instead of moving the paper beyond our remit we have clarified the title, abstract, and conclusion to more accurately reflect our limitations. We do of course still hope that our conclusions on connecting, engagement, environmental impact, and accessibility will be an inspiration for organisers of future meetings.

Of course, we would love to see this work used in collaboration with other surveys to draw a more holistic assessment of online conferencing during the pandemic but feel that it would not be appropriate for us to do so here with the limited data we have.

**Minor notes:**

*Suggest inserting '(SIC)' in the tables as well as the quotes in the main text, where appropriate.*

This is a good point and we have done so.

*Line 17: suggest removing 'of its size' as you have not yet described the size of the conference.*

This is a good point and we have done so.

*Line 21: suggest writing themes in italic or using another way of differentiating them from the rest of the text. Suggest doing the same in the rest of the paper.*

This is a good point and we have done so.

*Line 22: suggest rewording and starting a new sentence from where you say 'and include' to make sentence structure clearer.*

This section has been reworded for clarity.

*Line 48: change 'make' to 'made' to account for new year.*

This is a good catch – thank you! – and has been changed.

*Line 149: 'among others': should this read 'among other things'?*

This has been corrected, thank you!

*Line 210: change ':' to ','.*

This has been corrected.

*Table 2: In the definition of 'lack of engagement', should read 'fewer opportunities'.*

This has been corrected, thank you!

*Line 522: Missing word "made" after "been".*

This has been corrected.

---

## Author Comment (AC2)

Reviewer 2 Response

*This is an interesting and topical paper which I enjoyed reading and reviewing. It will interest many readers, and is relevant to many, beyond geoscience.*

*My comments and suggestions mostly focus on helping the authors to open out the paper to an audience or readership outside of EGU, since I find it assumes implicit knowledge of the organisation, EGU, and how its annual conference runs.*

*There's also opportunity to draw on literature about, say, environmental impacts of events and internal conflicts around climate action (in general the paper is quite sparse in references).*

*I'm also keen that the paper is broadened out. The content of the article currently refers to 'the future of EGU conferences' not 'the future of conferences'. Linking the discussion/analysis into how conferences are generally run (now and into the future) would broaden the content to wider readership. At the moment, my feeling is that a reader unfamiliar with EGU but keen to learn more about the 'future of conferences' might become lost.*

Dear Jen,

Thank you for this feedback to our paper, we are really pleased that you enjoyed it and saw the relevance of the work. We also appreciate the constructive feedback you provided, and we will address your main suggestions here and then the additional notes below. Firstly, with regard to broadening the scope of the paper to an audience beyond EGU this is good advice and we have made several changes that we hope will help with this suggestion. The other part of your suggestion which would expand the paper's applicability to conferences outside EGU we are reluctant to do as we are very limited by the scope and style of data we have access to, and don't want to broaden the paper beyond those specific limitations. In order to help clarify this limitation we have altered the title, abstract, and conclusion to clarify our intention, and to express our hope that this data can be combined with additional research from other conferences to draw the broader conclusions that you suggest.

**Other general comments:**

*There is the assumption that the reader knows the distinction between accessibility, inclusion, and diversity. I suggest to include definition or examples of what is meant be these measures.*

We have clarified the text around our definitions of these terms.

*There is also the assumption that the reader is familiar with EGU, and I think this assumption needs addressing. For example, replace reference to 'medal lectures' to 'awards ceremony' or something more generic.*

This is a good point and we have amended the text to reflect this.

*I felt the paper lacked references to peer reviewed literature on inclusion, accessibility, and environmental footprint of conferences and events. To provide evidenced rationale for why EGU are implementing measures to improve on these topics, and the responsibility of organisations such as EGU to improve.*

This is a reasonable point and we have added additional literature into the paper.

*There may be further future coronavirus pandemics and outbreaks, so I think the authors should specify that where they refer to coronavirus pandemic, they mean the COVID-19 disease pandemic, and possibly specify the virus itself (SARS-CoV-2)*

This has been corrected, thank you.

*I'd like to see a bit more data on how prevalent these codes were in the range of responses for different groups. Whether shown graphically or described. Particularly because for example (seemingly) one comment about concerns about open publishing of preliminary results (which I do very much understand concerns about) becomes one of four conclusions. I don't currently readily get a sense of which codes are the majority and by which group. And whether other indicators (gender, discipline etc) saw any differences in the responses. I'd like the results to be expanded in this way.*

We have added a graph to demonstrate the prevalence of codes in the sampled data by career stage into the paper, thank you for the recommendation.

*The discussion and conclusion on the (reduced) ability to connect and interact could be linked back to technological aspects which were removed from the analysis (for robust reasons). But perhaps the technological constraints affected this ability to connect and interact. Might the conference, if run differently, offer approaches to facilitate this aspect of conferences, for example through virtual networking lounges? How much of these effects are because of the timing (very soon after covid staled much travel in Europe, and adjustements to online working/networking) and expectation (expected face to face event in Vienna, pivoted to online). You do draw on this in the conclusions, but I feel this should go in the analysis/discussion section, too.*

This is possible and a very interesting idea but lies outside the scope of the data that we were able to collect at the time, and so we would not like to speculate on this further.

**Specific comments**

*Line 16 "six week pivot" is confusing. Suggest to clarify.*

Clarified, thank you.

*Line 26 – are they restrictive to positive and negative? Are some points neutral?*

This line has been clarified to show that this is range of opinions.

*Lines ~37 – I think it would be helpful here if the authors explicitly lay clear that EGU is a global network and a global conference. Else the reader can easily continue with the impression that*

*EGU = Europe only. This matters for later in the paper when noting wish to be inclusive of time zones etc.*

This has been clarified.

*Line 38 – spring is relative to where you are in the world. Suggest to specify the month.*

This has been specified.

*Line 48 – might you define early career scientist?*

We did not add a definition of Early Career Scientist, despite EGU having a definition for it, because for the purposes of the survey we asked participants to self-define (this is explained in section 2.2). This means we cannot impose a definition.

*Line 50 – who is "EGU" in this case – do you mean the secretariate or similar, or the EGU community? suggest to replace 'build up' with alternative wording such as 'formal planning'; for some, EGU 2021 would have been building up for many years!*

This has been clarified, thank you!

*Line 53 – "to lie on" is strange wording. Suggest rephrase.*

This has been rephrased.

*Line 59 - catering options can be an accessibility measure or an inclusion measure, depending on the measure.*

This has been adjusted for clarity.

*Line 62 – 'travel' and 'catering' in themselves are not measures, can you specify what the environmental measures achieved. For example, encouraging/incentivising delegates' sustainable travel choices to and within Vienna, or reducing catering waste by XYZ or reducing carbon intensity of catering options by widening plant based options. Are there sources of information (e.g. EGU reports, papers) that you could reference about these measures? You refer to environmental measures later in the paper (lines 504), whereas I think that text should go here.*

We think this text is valuable at the beginning of this section to add context, however we have also added more specific information in the section from line 535 onwards, to clarify this point.

*Paragraph starting Line 103*

> *You start off saying the live text chat was chosen as a compromise between accessibility and interaction. It might help here to lay out what the alternative options were, and perhaps also how these differed in terms of accessibility and interaction.*
>
> *The 'display' is well described, but I think that the text-chat and how it worked and was managed needs expanding on to make sense for readers who didn't attend EGU20. For example, please describe what the live text intended to be used for? Was it intended to*

*be used to ask questions, give comments, other? And how did the live text work? How long did it last, was the session timed to allow all those with 'displays' that were attending in the live text session to have turn in engaging, could any participant type into the chat, to be read by all, were there moderators? I know some of this from attending/co-convening, but other readers won't.*

Additional information on the methodology of the meeting, specifically how the text chat functioned and was moderated, has been added here.

*Line 122 – what does enabling Division meetings via 'Chat' mean?*

This has been clarified.

*Line 125 – what are the medal lectures? Would awards ceremony or session work instead? Quite a bit of this section assumes that the reader knows the EGU well, and I think the language needs broadening out. For example, you might not need to refer to Union Symposia and Great Debates – simply "Union-wide events" (as opposed to Division events).*

This is a good point and we have adjusted this throughout the paper. Thanks!

*Line 128 – really long sentence, suggest to break up.*

This has been fixed.

*Line 144 – mentoring programme isn't mentioned previously – was this done online during EGU20 or one of the activities that was cancelled?*

More information has been added here.

*Line 148 – extra support from who?*

This has been clarified.

*Line 152 – "this unique opportunity" lacks specificity. Refer directly to e.g. "The EGU20 online format offered unique opportunity"*

This has been clarified.

*Line 155 "this year" lacks specificity. Refer directly to the 2020 conference.*

This has been clarified.

*Lines 156 onwards are part of the methodology, and so could be moved forward.*

Whilst we see how this could be moved, we think this section needs to remain in place to give appropriate context for the methodology.

*Lines 221 – how are these career stages defined? This matters given your findings, later on.*

As described in the text, the participants were allowed to self-identify their career stage. Because of this we cannot apply definitions that may not apply to the participants.

*Line 487 – you note personal attitude, but I would caution that any attitude that one held in April/May 2020 might have changed given that over the past year+ EGU members have become much more acquainted with online conferences and events.*

This is a reasonable point and so we have adjusted the text.

*Lines 497 – from reduced footprint from travel, explicitly, or from broader environmental ramifications e.g. conference memorabilia, hotel stays, food and catering waste, other disposable items. My feeling would be to remove specific reference to travel, as you have raised this in the introduction.*

We have clarified this section whilst moving some text onwards from the introduction.

*Line 523 – what is this "start" and how would bikes be "promoted"? Suggest to be more specific in these measures.*

This section has been clarified.

*Lines 687 – I agree, but this isn't really drawn upon to the depths that it could be in the previous section. Is there content in your analysis that suggests negative impacts and solutions or suggestions? And how these compare to face-to-face conferences which have accessibility challenges too?*

Although we would like to delve deeper into this question our data doesn't provide enough scope to challenge these ideas, though other results from similar conferences may be able to provide better context in the future.